# Anatomical Site, Typing, Virulence Gene Profiling, Antimicrobial Susceptibility and Resistance Genes of *Streptococcus suis* Isolates Recovered from Pigs in Spain

**DOI:** 10.3390/antibiotics10060707

**Published:** 2021-06-11

**Authors:** Máximo Petrocchi-Rilo, Sonia Martínez-Martínez, Álvaro Aguarón-Turrientes, Elisabet Roca-Martínez, María-José García-Iglesias, Esther Pérez-Fernández, Alba González-Fernández, Elena Herencia-Lagunar, César-Bernardo Gutiérrez-Martín

**Affiliations:** 1Departmento de Sanidad Animal, Facultad de Veterinaria, Universidad de León, Campus de Vegazana s/n, 24071 León, Spain; mpetr@unileon.es (M.P.-R.); smarm@unileon.es (S.M.-M.); mjgari@unileon.es (M.-J.G.-I.); eperf@unileon.es (E.P.-F.); agonzf17@estudiantes.unileon.es (A.G.-F.); eherl00@estudiantes.unileon.es (E.H.-L.); 2Laboratorios SYVA, Avda. de Portugal s/n, 24009 León, Spain; alvaro.aguaron@syva.es (Á.A.-T.); e.roca@syva.es (E.R.-M.)

**Keywords:** *Streptococcus suis*, swine, typing, virulence genes, antimicrobial susceptibility, resistance genes

## Abstract

A set of 207 *Streptococcus suis* isolates were collected from ten autonomous communities from Spain in 2019 to 2020 from pigs with meningitis, pneumonic lungs, arthritic joints or other swollen viscera, to a lesser extent. Thirteen capsular types were detected being the most prevalent serotype 2 (21.7%), followed by serotypes 1 (21.3%), 9 (19.3%) and 3 (6.3%). Serotypes 2 and 9 were recovered mainly from the central nervous system (CNS), while serotype 1 was isolated mostly from swollen joints and serotype 3 from the lungs. Twenty-five isolates (12.1%) could not be typed. The most prevalent pathotype was *epf* + *mrp* + *sly* + *luxS* (49 isolates, 23.8%), and it was related mainly to serotypes 1 and 2. Serotypes 1–3 and 9 were significantly associated with anatomical sites of isolation and virulence factors, serotype 9 (CNS) and serotypes 3 and 9 (lungs) being associated with virulence profiles without the *epf* gene. *S. suis* isolates showed globally high antimicrobial resistances, but ampicillin followed by spectinomycin and tiamulin resulted in the highest activities, while the greatest resistances were detected for sulphadimethoxine, tetracyclines, neomycin, clindamycin and macrolides. A total of 87.4% isolates were positive to the *tetO* gene, 62.4% to the *ermB* gene and 25.2% to the *fexA* gene, while 14.6% were positive to all three genes simultaneously. A significative association between isolate resistances to tetracyclines and macrolides and the resistance genes tested was established, except for phenicol resistance and the *fexA* gene. A set of 14 multiresistance patterns were obtained according to the number of antimicrobials to which the isolates were resistant, the resistances to 12 or more agents being the most prevalent ones. A remarkable amount of multiresistance profiles could be seen among the *S. suis* serotype 9 isolates.

## 1. Introduction

The porcine respiratory disease complex (PRDC) is one of the most important health concerns affecting swine production, resulting in poor growth rates and high management costs [1]. *Streptococcus suis* is one of the secondary infectious agents frequently detected in PRDC and the only Gram-positive organism currently recognized within this complex. Separately, *S. suis* is responsible for a variety of infections such as meningitis, septicemia, arthritis, endocarditis or bronchopneumonia [2]. However, clinically healthy pigs can also carry these organisms in the tonsils and upper respiratory tract, which is a noticeable way for them to spread. Along with the economic losses caused in pig husbandry, the severity of *S. suis* relies upon its condition of being a zoonotic agent responsible for sporadic human cases of meningitis or septicemia [3].

So far, 35 *S. suis* serotypes based on capsular polysaccharides have been reported [4]; however, several serotypes, such as 20, 22, 26 and 32–34, should be reclassified as belonging to other species of the genus *Streptococcus* on the basis of molecular approaches [5,6]. Serotype 2 isolates are the most prevalent worldwide and, together with serotype 9, are the most virulent; in addition, this latter serotype has been commonly isolated from sick pigs in many European countries [7].

*S. suis* virulence varies depending on the serotypes and clinical isolates within the serotypes. The virulence of serotypes 1 and 2 is linked to three proteins: the muramidase-released-protein (MRP), extracellular-factor protein (EF) and suilysin. Thus, isolates positive for MRP and EF are virulent, while those negative for these two proteins are considered less virulent or even nonvirulent, and they are recovered usually from the tonsils of healthy swine [8]. Nevertheless, MRP and EF are not crucial, and, for instance, most of the isolates belonging to serotype 2 recovered in Canada do not generate these two proteins [9]; however, the majority of those isolated in some European countries express them [10]. EF has been also related to virulence in serotypes 1/2 and 14; by contrast, EF expression is not associated with virulence in serotypes 7 and 9, while the expression of a MRP variant seems to be linked to virulence in serotype 9 [11]. In addition, a subtilisin-like protease [12], an oligopeptide-binding protein [13] and a parvulin-type peptidyl-prolyl isomerase [14] have been described in isolates belonging to serotype 2.

The use of antimicrobial agents is necessary to control the bacteria entailed in PRDC, but their misuse is one of the factors involved in the emergence and spread of bacterial resistances across the world [15]. Although these compounds are important for the treatment and control of *S. suis* infections, a relatively high number of resistances have been reported during the last decades to the classes commonly used in swine, which vary according to countries, serotypes and over the years [16]. The concept of an antimicrobial resistome has been proposed for reporting the collection of all known antimicrobial resistance genes in the microbial ecosystem. This term supports the hypothesis that resistant isolates and their resistomes are settled after birth in piglets and are gained from the mother or by direct contact with resistant organisms in the surrounding environment [17].

One of the purposes of this study was to determine the molecular typing of *S. suis* isolates recovered from pigs in Spain during 2019 and 2020, as well as their virulence genes, in order to broaden the information about the epidemiology of the infections caused by *S. suis* in Spain in comparison with the findings from other countries. The other aims of this investigation were to ascertain the in vitro antimicrobial susceptibility, as well as resistotypes, and to characterize the potential presence of six resistance genes and their putative association with resistance.

## 2. Results

### 2.1. Typing of Streptococcus suis Isolates

A set of 207 *S. suis* isolates coming from 147 intensive pig farms located throughout Spain recovered from 2019 to 2020 were examined.

Most of the *S. suis* isolates (68.6%) belonged to serotypes 1 (21.3%), 2 (21.7%), 3 (6.3%) or 9 (19.3%), while 25 isolates (12.1%) were non-typable when using multiplex PCR. The remaining nine serotypes involved less than 5%. The distribution of the 13 serotypes and the non-typable isolates is shown in Table 1.

Taken into account the serotypes and anatomical site of isolation (Table 1), the clinical strains recovered from the CNS, lungs and joints added up 94.2% of the locations. Serotype 1 was the most prevalent in the samples coming from the joints (38.0%), followed from afar by serotypes 2 (20.7%) and 9 (19.0%). Besides, serotype 9 was the main serotype recovered from the CNS (23.4%), ahead of serotypes 1 and 2 (21.9% each). The statistical analysis carried out on the most frequent serotypes and their main anatomical origins showed that serotype 3 was recovered mostly from the lungs (92.3%) unlike serotypes 1, 2 or 9 (*p* < 0.01).

On the other hand, different serotypes were isolated from several places in the same pig; for instance, serotype 1 was recovered from the lungs and serotype 9 from the CNS. As another example, serotype 1 was isolated from the CNS, while serotype 5 was recovered from the joints. Furthermore, serotypes 1 and 9 were isolated from the lungs in the same animal. Even more, two isolates belonging to serotype 2 were collected from the same animal, one from the lungs and the other from the liver, but an interesting fact was that these two isolates were different, as was shown by different virulence profiles in each of them. The highest number of unlike serotypes found in one single pig was two.

### 2.2. Determination of Virulence-Associated Genes in Streptococcus suis Isolates

Taken one by one the five virulence genes tested, the most prevalent was *luxS* (in 92.8% of the *S. suis* isolates), followed by *mrp* (in 73.9%) and *sly* (in 67.6%). In addition, the *gapdh* and *epf* genes were located rather less, being positive in 44.4% and 44.0% of the isolates, respectively. Twenty-three profiles were found when the five genes were mixed, according to their different combinations (Table 2). Only four isolates (1.9%) lacked all of these genes: two were recovered from the lungs, another from the CNS, and the fourth from the joints. As an oddity, this virulence profile (absence of these genes) was related to non-typable serotypes in all cases. The most prevalent pattern (49 isolates, 23.8%) showed the four virulence genes, except the *gapdh* gene, and mostly included isolates belonging to serotype 1 (59.2%) and then serotype 2 (34.7%).

Far away, the second-most frequent profile lacked only the *epf* gene (26 isolates, 12.7%), and half of them were assigned to serotype 9. Thereafter, 22 *S. suis* (10.6%) harbored *mrp* + *sly* + *luxS* genes, most also allocated to serotype 9 (63.6%), while 21 isolates (10.1%) retained all the five virulence genes. This latter pathotype incorporated isolates belonging mainly to serotypes 1 and 2. The remaining patterns were found in less than 10% of the cases. Of the clinical strains harboring only one virulence gene (but not the others), *luxS* can be highlighted, because it was detected in 14 isolates (6.8%), and half of them were non-typable (Table 2).

The assessment of virulence factors being present in the most frequent serotypes showed that the serotype 1 and 2 profiles differed significantly from those found in serotypes 3 and 9 (*p* < 0.01). Thus, the two most common patterns in serotypes 1 and 2 were *epf* + *mrp* + *sly* + *luxS* and *epf* + *mrp* + *sly* + *luxS* + *gapdh*, while these combinations were absent in serotypes 3 and 9. However, the main virulence profiles in serotypes 3 and 9 were *mrp* + *sly* + *luxS* and *mrp* + *sly* + *luxS* + *gapdh*. Concerning the isolation site regardless of serotypes, the virulence patterns differed significantly only between the joints and lungs (*p* < 0.05), mainly because of the absence of the *epf* + *mrp* + *sly* + *luxS* + *gapdh* profile in the lungs.

When the virulence factors of the most frequent serotypes were evaluated in the three main body sites, the statistical analysis revealed a similar pattern in the joints and CNS (Figure 1a,b). In both locations, the virulence profiles did not differ significantly between serotypes 1 and 2. However, these two serotypes differed significantly from those in serotype 9 (*p* < 0.01), in which *epf* + *mrp* + *sly* + *luxS* and *epf* + *mrp* + *sly* + *luxS* + *gapdh* combinations were absent (Figure 1a–c) while *mrp* + *sly* + *luxS* and *mrp* + *sly* + *luxS* + *gapdh* patterns were predominant, unlike serotypes 1 and 2. Likewise, serotypes 1 and 2 isolated from the lungs exhibited similar virulence patterns, while these two serotypes showed significant differences with respect to serotypes 3 and 9 (Figure 1c). Thus, the *epf* + *mrp* + *sly* + *luxS* profile prevailed in serotypes 1 and 2, while this pattern was absent in serotypes 3 and 9, which predominated those profiles in which the *epf* gene was missing.

### 2.3. Antimicrobial Susceptibility Testing in Streptococcus suis Isolates

The range, MIC_50_, MIC_90_ and rate of antimicrobial resistance of the 103 selected *S. suis* are shown in Table 3. These isolates showed the overall high resistances, because more than a half of them were resistant against nine of the antimicrobial agents tested—concretely, the two tetracyclines, neomycin, danofloxacin, clindamycin, sulphadimethoxine and the three macrolides (all with figures of a greater than 60% resistance). By contrast, the compound exhibiting the highest activity was ampicillin, followed by spectinomycin, with 97.1% and 88.4% of the isolates being susceptible, respectively. The two other β-lactams (ceftiofur and penicillin) also had good activity, with 82.5% and 73.8% of the isolates being susceptible, respectively.

Tiamulin had activity against 87.4% of the isolates, but a scarce effectiveness (5.8%) was found for sulphadimethoxine; however, a resistance to the combination of sulfamethoxazole + trimethoprim decreased considerably (until 34.9%) with regard to the other sulfonamide compared. Meanwhile, resistance levels under 20% were encountered in gentamicin and florfenicol but that of enrofloxacin rose to almost 50% of the isolates (Table 3).

A total of 65.1% of the *S. suis* isolates were resistant to a slot between 12 and 14 compounds and 98.0% were to at least seven antimicrobials. There was no isolate susceptible to all the 18 compounds evaluated, and only one (NT, isolated from the lungs) was susceptible to 15 antimicrobial agents, while other isolate (serotype 3, also recovered from the lungs) was susceptible to 14 compounds. On the opposite site, one clinical strain (serotype 7 from the CNS) was resistant to all of the antimicrobial agents tested. Serotype 9 isolates appeared mainly related to the multiresistance profiles (Table 4).

### 2.4. Detection of Resistance Genes

Of the six resistance genes examined, *tetM* was harbored by only one isolate (1.0%), recovered from the joints and ascribed to serotype 1, while 87.5% were positive to *tetO*, but none of the isolates showed *tetM* + *tetO* genes simultaneously. A significant association between tetracycline resistance genes and a resistance to tetracycline was found (*p* < 0.01; χ^2^ test).

On the other hand, 65.1% of the isolates amplified the *ermB* gene, but only one showed it in addition to the *mefA/E* gene (NT, recovered from the joints). All *S. suis* strains were resistant to macrolides when the *ermB* or *ermB* + *mefA/E* genes were present, and, for this reason, it was impossible to know whether *mefA/E* on its own produced a resistance to macrolides. A significant association between the *ermB* gene and resistance against macrolides was also demonstrated (*p* < 0.01; χ^2^ test). On the contrary, there was no significant association between the presence of the *fexA* gene and the resistance to florfenicol.

Furthermore, 43.6% of the strains showed exclusively the *tetO* + *ermB* linkage, while, with *fexA*, it was found alone or in combination with other genes in 26.2% of isolates, but none of them amplified the *cfr* gene. Twenty-one point five percent of the *S. suis* isolates showed at least the *tetO* + *fexA* genes, while 17.6% amplified at least the *ermB* + *fexA* genes. Finally, 15.6% were positive simultaneously to the *tetO* + *ermB* + *fexA* genes, 43.7% of which were ascribed to serotype 9 (Table 5).

## 3. Discussion

A collection of 207 *S. suis* genes recovered from all around Spain were tested for molecular typing, the main virulence genes, susceptibility to the common antimicrobial agents used for disorders in swine husbandry and the presence of six resistance genes in three antimicrobial classes.

The prevalence of serotypes 1 and 2 was similar to that previously described in Spain [2,18] and in other European countries [19]. However, Prieto et al. [18] found that the isolates of serotype 2 were almost six-fold higher than those ascribed to serotype 1, while among our 207 isolates, the percent of these two serotypes was practically the same. The considerably higher number of isolates belonging to capsular type 1 in our report with regard to other Spanish investigations carried out over a decade ago [2,20] seems to reveal a clear emergence of this serotype in the early years of the 21st century.

The majority of isolates of serotypes 2, 7 and 9 were associated with cases of pig meningitis in a previous study, while most of those belonging to serotype 3 were associated with pigs suffering pneumonia [21]. A similar tendency was observed in our investigation almost two decades after for the CNS and lung isolates, except for serotype 7 and the lungs, because only one strain belonging to serotype 7 was isolated from this location. On the other hand, the identification of one isolate classified as serotype 23 must be highlighted, because this serotype has not been described before in Europe to the best of our knowledge, in contrast to what happens in America (Minnesota) where it reached 10.0% [22] or in Asia, where serotype 23 was the most prevalent in Thailand [23].

Additionally, the percentage of non-typable isolates took fourth place among the serotypes in the present study, a finding quite in agreement with those described in Spain [2], Asia [24,25] or the United States [22]. However, it was substantially lower than the percent reported in Thailand, where non-typable isolates were the most prevalent ones [23,26,27]. Non-typable isolates could correspond to already described types with some mutations in the capsular biosynthesis genes or, alternatively, to unencapsulated clinical isolates [26]. Other hypothesis was based on the high genetic diversity of *S. suis* capsular polysaccharide loci [28].

Interestingly, almost half of the isolates showed at least four of the five virulence genes tested, and most belonged to serotypes 1, 2 or 9, thus emphasizing the virulence of these three serotypes among the 35 serotypes described so far. This higher virulence of these serotypes in relation to the remaining ones was previously reported [29]. The linkage between *epf* + *mrp* or *epf* + *mrp* + *sly* pathotypes and capsular types 1 and 2 was previously documented [2,25,30]. In a recent study conducted in Brazil [31], the simultaneous presence of *epf* + *mrp* + *sly* genes was shown in 47.9% of isolates, fourteen percent points above the value found for the Spanish *S. suis* isolates.

The fact that 8.7% of strains harbored exclusively one gene could indicate that only one of these virulence determinants would be enough to cause disease among the pigs sampled by us, a conclusion already obtained by Oh et al. [25]. However, Wongsawan et al. [26] reported a high prevalence of only the *mrp* or *sly* genotypes among the *S. suis* isolates recovered from pig carcasses in some wet markets in Thailand, which seems to indicate a lower virulence, unlike those in our study. On the other hand, a low percentage (1.9%) of the isolates without any of the five virulence genes was detected in our study, compared to 24.6% also devoid of the virulence genes recorded in Korea [25].

Regarding antimicrobial susceptibility studies, β-lactams have, over the years, been the antibiotics of choice for the treatment of disorders caused by *S. suis*. Even so, scarce resistances have been developed, and most isolates have proven to be susceptible in many researches [21,25,27,30,31,32]. Of the three β-lactams compared, ampicillin showed the best activity, in accordance with that found by Zhang et al. [33]. A correct use of antimicrobial agents against *S. suis* (as against any other bacterial pathogen) becomes critical to keep the therapeutic efficacy of the broad-spectrum antibiotics and to minimize the recruitment of resistances. For this reason, the values detected in our study, with rates for penicillin over one-fourth among the isolates tested and only a scarce percent against ampicillin, do not advise penicillin (but ampicillin) as one of the best options for the treatment of *S. suis* infections at present in Spain. The emergence of resistances for penicillin, gentamicin and enrofloxacin among the *S. suis* isolates is predicted in the near future in Thailand [34]. Our results, exhibiting resistances above 16% for these three compounds, already seem to evince this fact at this moment in Spain.

As in this report, a high resistance to neomycin was detected among the Spanish isolates recovered from the tonsils of slaughtered pigs twenty-five years ago [35]. Similarly, the resistance to spectinomycin herein was quite in agreement with that reported for the other isolates [21,30]; nevertheless, the resistance to spectinomycin observed in other investigation [31] was almost four times the value detected here. On the other hand, considerably lower MIC_90_ values than those found by us have been reported recently in the USA [36] for enrofloxacin (0.25 vs. 2 μg/mL)**,** florfenicol (4 vs. 8 μg/mL), tilmicosin (≥8 vs. >64 μg/mL) and tulathromycin (≥8 vs. >64 μg/mL). As in our results, widespread resistances to macrolides were also shown previously [25,30,32,34,37]. This outcome could easily be explained by the broad use of these antibiotics for the treatment of swine diseases, which could have enabled a selective pressure for macrolide-resistant *S. suis* isolates to grow and spread in pig farming.

High susceptibilities were detected for fluoroquinolones in several previous reports [21,27,30,36,38], which does not fit with our findings. The resistances to these antimicrobial agents were linked to single-point mutations in the quinolone resistance-determining regions of the *gyrA* and *parC* genes in other Spanish isolates [39]. Concerning phenicols, our rate of resistance to florfenicol was substantially higher than that found in other countries [31,34,38,40,41] but rather resembled that described in Brazil [42]. The existence of 15 isolates showing a lack of susceptibility to florfenicol could be explained because of its enhanced use in Spain after chloramphenicol was banned in food animals. However, this resistance could not be related to the presence of the *fexA* gene; therefore, it must rely upon other alternative mechanisms, such as the expression of the *cat* gene, a finding reported lately for chloramphenicol [43]. Additionally, this resistance could not be linked to the *cfr* gene, because the detection of this gene was negative for all the isolates in our study.

The extremely high resistance obtained to chlortetracycline and oxytetracycline was in general agreement with previous reports about other isolates [25,27,30,31,32,37,38,40] and especially matched with the rate over 90% also found in Spain 15 years ago [21]. Such a great resistance is undoubtedly related to their widespread use in swine production, and it could be directly associated in our study with *tetM* but, mainly, with the *tetO* gene. Unlike this report, either similar prevalences of *tetM* and *tetO* [44] or a dominant presence of *tetM* gene was already recorded in *S. suis* strains [45]. In addition, a resistance to tetracyclines is an important cofactor in the selection of resistances to macrolides [33,46], a feature also seen in our study that could explain the high resistances detected for tylosin, tilmicosin or tulathromycin and which could be related to the *ermB* gene. Similarly, macrolide resistances were associated with *ermB* but not with the *mef(A/E)* gene [47].

Slightly higher resistances for clindamycin compared to those in our work were aforementioned [24,42], while the MIC range seen by Werinder et al. [27] was quite similar to the one reported in this study. However, several authors proved lower values [30,32,40]. In contrast to the great resistances found recently for tiamulin [31,34], a high susceptibility was recorded for this pleuromutilin among the Spanish isolates tested by us. Besides, these latter isolates showed a MIC range quite similar to that by Zhang et al. [33]. The level of resistance shown for sulfamethoxazole/trimethoprim was approximately double that encountered among Asian isolates [32]. Even more different, this combination was proposed as an alternative drug in the treatment of *S. suis* infections in Spain almost three decades ago [20]. On the opposite side, other Spanish clinical strains [35] resulted in resistances to sulfamethoxazole/trimethoprim almost double to that in our investigation.

A high diversity of profiles was found among the resistotypes obtained, a finding similar to that already reported [18,34,42]. However, in other recent studies [31,32], the most common patterns included only five or six agents at most, while a multiple resistance to nine compounds was the dominant phenotype in another investigation [24]. A simultaneous resistance to multiple antibiotic classes, such as to tetracyclines, macrolides and aminoglycosides (also detected in our report), has been related to the integrative conjugative elements being passed through horizontal gene transfer [34,48].

## 4. Materials and Methods

### 4.1. Streptococcus suis Isolates, DNA Extraction and Molecular Typing

A total of 207 *S. suis* isolates coming from 147 intensive pig farms located in ten Spanish autonomous communities (Andalucía, Aragón, Castilla-La Mancha, Castilla y León, Cataluña, Comunidad Valenciana, Extremadura, Galicia, Madrid and Murcia) and collected between April 2019 and June 2020 were tested. Samples were taken from the central nervous system (CNS), joints, lungs, liver and/or other viscera but only when they showed lesions suspicious of the infection caused by *S. suis*. In detail, 73 isolates were recovered from brains with meningitis, 64 from pneumonic lungs, 58 from joints with arthritis, one from a swollen liver and 11 from other swollen viscera (Table 1). They were isolated from these samples on Columbia blood agar plates (Oxoid Ltd., Madrid, Spain) containing 5% defibrinated sheep blood after 48 h at 37 °C under aerobic conditions.

The identification of presumptive isolates was carried out by standard procedures (colony morphology, α-hemolytic activity, Gram-positive staining and catalase negative activity), and then, they were confirmed by a PCR based on the glutamate dehydrogenase gene [49]. For DNA extraction, about ten *S. suis* colonies from chocolate agar cultures were resuspended in 100 μL of deionized water and maintained at boiling temperature for ten min. Then, they were centrifuged at 13,000× *g* for 10 min, and the supernatant was kept at −20 °C until use. The primers used were JP4 (5′-GCAGCGTATTCTGTCAAACG-3′) and JP5 (5′-CCATGGACAGATAAAGATGG-3′). The PCR assay comprised 5 min of preincubation at 94 °C, followed by 35 cycles of 1 min of denaturation at 94 °C, 1 min of annealing at 55 °C and 1 min of extension at 72 °C. The final extension was performed for 7 min at 72 °C.

All isolates were typed using a multiplex PCR, which was carried out independently in four sets [50]. The first set included the primers for serotypes 1/2, 1, 2, 3, 7, 9, 11, 14 and 16; the second for serotypes 4, 5, 8, 12, 18, 19, 24 and 25; the third for serotypes 6, 10, 13, 15, 17, 23 and 31 and the fourth for serotypes 21, 27, 28, 29 and 30 [29,51,52,53]. About ten *S. suis* colonies were resuspended in 100 μL MilliQ water. This suspension was boiled at 100 °C for 10 min, and then centrifuged at 12,000× *g* for 10 min. The supernatant was collected and stored at −20 °C until use. The PCR mixture was described by Kerdsin et al. [50], and the thermal profile consisted of an initial activation at 95 °C for 3 min, 30 cycles of denaturation at 95 °C for 20 s, annealing and extension at 62 °C for 90 s and a final extension at 72 °C for 5 min.

### 4.2. Detection of Virulence Genes in Streptococcus suis Isolates

The 207 *S. suis* isolates were further investigated for virulence-associated factors by PCR according to the previous protocols. The five genes tested were *mrp*—codifying the muramidase-released protein [29], *epf*—extracellular factor protein [29], *sly*—suilysin [51], *luxS*—S-ribosylhomocysteinase [52] and *gapdh*—glyceraldehyde-3-phosphate dehydrogenase [53].

### 4.3. Antimicrobial Sensitivity Testing

A selection of half of the *S. suis* isolates (*n* = 103) was chosen proportionally to the serotypes found in this study. The antimicrobial susceptibilities were determined by the broth microdilution method by means of a commercially prepared, dehydrated 96-well microtiter plates (BOPO6F, Sensititer; Trek Diagnostic Systems Inc., East Grinsted, UK) in accordance with the recommendations by the Clinical and Laboratory Standards Institute [54,55]. Müeller–Hinton broth with 5% of sterile fetal calf serum was used as the broth medium, and then, the panel was reconstituted by adding 100 μL/well of the inoculum, and plates were incubated at 37 ± 2 °C for 18–24 h. The antimicrobial agents used and their dilution ranges are shown in Table 3. Each panel was read visually, and the minimal inhibition concentration (MIC) was established as the lowest concentration of antimicrobial agent inhibiting visible growth of the 100 μL/well-inoculated agents.

### 4.4. Detection of Antimicrobial Resistance Genes

Six antimicrobial resistance genes belonging to three different classes (tetracyclines—*tetM* and *tetO*, macrolides—*ermB* and *mefA/E* and phenicols—*cfr* and *fexA*) were examined by PCR for the 103 isolates examined in Section 4.3, according to previous reports [46,56,57].

### 4.5. Statistical Analysis

The association among the most prevalent serotypes (1, 2, 3 and 9); the main anatomical sites of isolation (joints, central nervous system (CNS) and lungs) and the four most frequent virulence profiles in *S. suis* was tested by the χ^2^ test for independence, and the data were expressed as percentages. In addition, this same test was used to determine the association between the isolates harboring, or not, a resistance gene of the tetracycline, macrolide or phenicol classes and the susceptibility or resistance of each isolate. SPSS software version 24 (SPSS Inc., IBM, Chicago, IL, USA) was used to carry out the statistical analysis. Significant differences were considered for *p* < 0.05.

## 5. Conclusions

A significant association could be established between the virulence profiles of the most prevalent *S. suis* serotypes (1–3 and 9) and the main anatomical sites from which they were recovered from. These isolates carried three or more virulence factors, and most harbored multiresistance patterns. The overall resistance (associated with the *tetO* and/or *tetM* genes for tetracyclines and with the *ermB* genes for macrolides) was probably because of the great selective pressure exerted and of their misuse. Therefore, a prudent use of these compounds in therapy for infections caused by *S. suis* and the need for the continuous surveillance of resistance patterns are strongly advised.

## Figures and Tables

**Figure 1 antibiotics-10-00707-f001:**
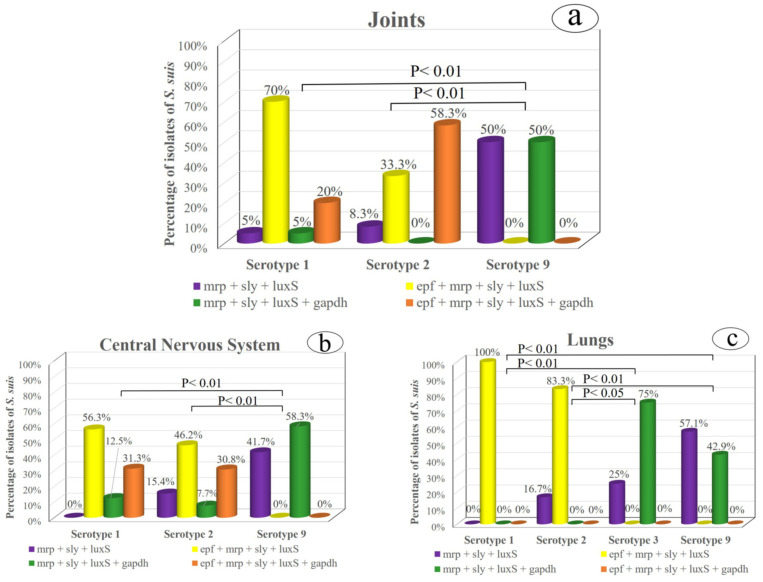
Relationship between the most prevalent *S. suis* serotypes and their virulence factors in the three main isolation origins: (**a**) the joints, (**b**) central nervous system and (**c**) lungs.

**Table 1 antibiotics-10-00707-t001:** Distribution of *Streptococcus suis* isolates according to serotype and anatomical origin.

Serotype	Number of Farms Tested	Place of Isolation (Number and Percentage)					Total ***
		Joints	Central Nervous System (CNS)	Lungs	Liver	Other Viscera	
1	26	22 (50.0%) * (38.0%) **	16 (36.4%) * (21.9%) **	6 (13.6%) * (9.4%) **	0	0	44 (21.3%)
2	30	12 (26.7%) * (20.7%) **	16 (35.6%) * (21.9%) **	14 (31.1%) * (22.0%) **	1 (2.2%) * (100%) **	2 (4.4%) * (18.2%) **	45 (21.7%)
3	13	0	1 (7.7%) * (1.4%) **	12 (92.3%) * (18.7%) **	0	0	13 (6.3%)
4	6	0	2 (28.6%) * (2.7%) **	4 (57.1%) * (6.2%) **	0	1 (14.3%) * (9.1%) **	7 (3.4%)
5	2	1 (50.0%) * (1.7%) **	1 (50.0%) * (1.4%) **	0	0	0	2 (1.0%)
7	5	3 (42.8%) * (5.2%) **	2 (28.6%) * (2.7%) **	1 (14.3%) * (1.6%) **	0	1 (14.3%) * (9.1%) **	7 (3.4%)
8	6	0	0	8 (88.9%) * (12.5%) **	0	1 (1.1%) * (9.1%) **	9 (4.3%)
9	27	11 (27.5%) * (19.0) **	17 (42.5%) * (23.4%) **	10 (25.0%) * (15.6%) **	0	2 (5.0%) * (18.2%) **	40 (19.3%)
10	1	0	1 (100%) * (1.4%) **	0	0	0	1 (0.5%)
16	5	2 (40.0%) * (3.4%) **	3 (60.0%) * (4.1%) **	0	0	0	5 (2.4%)
17	2	0	2 (66.7%) * (2.7%) **	0	0	1 (33.3%) * (9.1%) **	3 (1.4%)
19	3	1 (25.0%) * (1.7%) **	2 (50.0%) * (2.7%) **	0	0	1 (25.0%) * (9.1%) **	4 (1.9%)
23	2	0	0	2 (100%) * (3.1%) **	0	0	2 (1.0%)
Non-typable	19	6 (24.0%) * (10.3%) **	10 (40.0%) * (13.7%) **	7 (28.0%) * (10.9%) **	0	2 (8.0%) * (18.2%) **	25 (12.1%)
Total ***	147	58 (28.0%)	73 (35.3%)	64 (30.9%)	1 (0.5%)	11 (5.3%)	207 (100%)

* Percentage of the isolates belonging to this serotype. ** Percentage of the isolates recovered from this anatomical site. *** Percentage of the total of the isolates.

**Table 2 antibiotics-10-00707-t002:** Virulence gene profiles of the 207 *Streptococcus suis* genes tested in this study according to serotypes.

Virulence Gene	Number of Isolates (%)	Serotypes
none	4 (1.9)	NT (*n* = 4)
*epf*	1 (0.5)	NT
*luxS*	14 (6.8)	3 *, 7, 9, 17, 19 (*n* = 3), NT (*n* = 7)
*gapdh*	3 (1.4)	NT (*n* = 3)
*epf* + *mrp*	1 (0.5)	NT
*epf* + *sly*	3 (1.4)	16 (*n* = 2), 17
*epf* + *gapdh*	1 (0.5)	NT
*mrp* + *luxS*	11 (5.3)	2, 3, 7, 8, 9 (*n* = 5), 23, NT
*sly* + *luxS*	3 (1.4)	4, 8, 9
*sly* + *gapdh*	1 (0.5)	3
*luxS* + *gapdh*	5 (2.4)	1, 5, 17, NT (*n* = 2)
*epf* + *mrp* + *luxS*	3 (1.4)	1, 2 (*n* = 2)
*epf* + *sly* + *luxS*	6 (2.9)	2 (*n* = 4), 8, NT
*epf* + *luxS* + *gapdh*	1 (0.5)	10
*mrp* + *sly* + *luxS*	22 (10.6)	1, 2 (*n* = 4), 3, 8, 9 (*n* = 14), NT
*mrp* + *sly* + *gadph*	1 (0.5)	2
*mrp* + *luxS* + *gadph*	17 (8.2)	2 (*n* = 2), 3 (*n* = 3), 5, 7 (*n* = 4), 9 (*n* = 5), 16, NT
*sly* + *luxS* + *gapdh*	9 (4.3)	3 (*n* = 2), 8 (*n* = 4), 7, 16, 23
*epf* + *mrp* + *sly* + *luxS*	49 (23.8)	1 (*n* = 29), 2 (*n* = 17), 4 (*n* = 2), NT
*epf* + *mrp* + *luxS* + *gapdh*	2 (1.0)	19
*epf* + *sly* + *luxS* + *gapdh*	3 (1.4)	2, 9, NT
*mrp* + *sly* + *luxS* + *gapdh*	26 (12.7)	1 (*n* = 3), 2 (*n* = 2), 3 (*n* = 3), 4 (*n* = 3), 8, 9 (*n* = 13), 16
*epf* + *mrp* + *sly* + *luxS* + *gapdh*	21 (10.1)	1 (*n* = 9), 2 (*n* = 11), 4

* If “*n*” does not appear, it means that *n* = 1.

**Table 3 antibiotics-10-00707-t003:** MIC (minimum inhibitory concentration) range, MIC_50_, MIC_90_ and percentage of resistance of 103 *Streptococcus suis* isolates.

Antimicrobial Agent (Dilution Range)	Range (μg/mL)	MIC_50_ (μg/mL)	MIC_90_ (μg/mL)	Breakpoint (μg/mL) *	Antimicrobial Resistance (%)
Penicillin (0.12–8 μg/mL)	0.12–>8	0.12	2.0	1	26.2
Ampicillin (0.25–16 μg/mL)	0.25–>16	0.25	0.5	1	2.9
Ceftiofur (0.25–8 μg/mL)	0.25–>32	0.25	8.0	4	17.5
Chlortetracycline (0.5–8 μg/mL)	0.5–>8	>8	>8	4	93.2
Oxytetracycline (0.5–8 μg/mL)	0.5–>8	>8	>8	4	93.2
Gentamicin (1–16 μg/mL)	2–>16	8	16	8	16.5
Neomycin (4–32 μg/mL)	4–>32	32	>32	8	88.3
Spectinomycin (8–64 μg/mL)	16–>64	32	>64	64	11.6
Enrofloxacin (0.12–2 μg/mL)	0.5–>2	1	2	1	46.6
Danofloxacin (0.12–1 μg/mL)	0.12–>1	>1	>1	1	61.2
Florfenicol (0.25–8 μg/mL)	2–8	4	8	4	14.6
Clindamycin (0.25–16 μg/mL)	0.25–>32	>32	>32	1	87.4
Sulfamethoxazole/trimethoprim (38/2 μg/mL)	38/2–>38/2	38/2	>38/2	38/2	34.9
Sulphadimethoxine (256 μg/mL)	256–>256	>256	>256	256	94.2
Tiamulin (0.5–32 μg/mL)	0.5–>32	4	>32	32	12.6
Tylosin (0.5–32 μg/mL)	0.5–>32	>32	>32	32	86.4
Tilmicosin (4–64 μg/mL)	4–>64	>64	>64	32	84.5
Tulathromycin (1–64 μg/mL)	2–>64	>64	>64	32	85.4

* Clinical breakpoints (CLSI VET08 (2018a), CLSI M100 (2018b), Vela et al. (2005), Li et al. (2012), Van Hout et al. (2016), Niemann et al. (2018), Ichikawa et al. (2019) or Werinder et al. (2020)).

**Table 4 antibiotics-10-00707-t004:** Antimicrobial resistance profiles among the 103 *Streptococcus suis* isolates tested, according to the number of antimicrobial agents to which they showed resistance and to the serotypes to which they belonged.

Number of Antimicrobial Agents to Which Resistance Was Detected	Number of Isolates (%)	Serotypes
Three *	1 (1.0)	NT
Four	1 (1.0)	3
Seven	3 (2.9)	1 (*n* = 2), NT **
Eight	2 (1.9)	1, 2
Nine	5 (4.8)	1, 2 (*n* = 3), 10
Ten	3 (2.9)	1, 2 (*n* = 2)
Eleven	4 (3.9)	2, 3 (*n* = 2), 9
Twelve	22 (21.4)	1 (*n* = 2), 2 (*n* = 4), 3, 4, 5 7, 9 (*n* = 5), 16 (*n* = 2) 23, NT (*n* = 2)
Thirteen	29 (28.2)	1 (*n* = 9), 2 (*n* = 7), 3, 7 (*n* = 2) 9 (*n* = 6), 19, NT (*n* = 2)
Fourteen	16 (15.5)	1 (*n* = 2), 2 (*n* = 3), 3, 4, 8 9 (*n* = 2), NT (*n* = 3)
Fifteen	6 (5.8)	1, 4, 8, 9, 17, NT
Sixteen	7 (6.8)	1, 2, 9 (*n* = 5)
Seventeen	3 (2.9)	7, 9, NT
Eighteen (all the antimicrobial agents tested)	1 (1.0)	7

* The antimicrobials to which resistances were detected cannot be indicated, because the number of them (three, four…, seventeen) did not always cover the same compound and the same class. ** If “*n*” does not appear, it means that *n* = 1.

**Table 5 antibiotics-10-00707-t005:** Resistance genes found (*tetM*, *tetO*, *ermB*, *mefA/E*, *cfr* and *fexA*) in the 103 *Streptococcus suis* tested and the serotypes to which they belonged.

Resistance Gene	Number of Isolates (%)	Serotypes
none	5 (4.9)	1 (*n* = 2), 3 (*n* = 2), 8 *
*tetO*	22 (21.4)	
*ermB*	2 (1.9)	9, NT
*fexA*	3 (2.7)	1 (*n* = 3)
*tetM* + *ermB*	1 (1.0)	1
*tetO* + *ermB*	45 (43.6)	1 (*n* = 14), 2 (*n* = 7), 3 (*n* = 4), 4 (*n* = 2), 5, 7 (*n* = 3), 8, 9 (*n* = 7), 19, 23, NT (*n* = 4)
*tetO* + *fexA*	6 (5.9)	2 (*n* = 3), 7, 9 (*n* = 2)
*ermB* + *fexA*	1 (1.0)	9
*tetM* + *ermB* + *fexA*	1 (1.0)	1
*tetO* + *ermB* + *mefA/E*	1 (1.0)	NT
*tetO* + *ermB* + *fexA*	16 (15.6)	2 (*n* = 2), 3, 4, 8 (*n* = 2) 9 (*n* = 7), 16, NT (*n* = 2)

* If “*n*” does not appear, it means that *n* = 1.

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
