# Peer review of "Anatomical Site, Typing, Virulence Gene Profiling, Antimicrobial Susceptibility and Resistance Genes of Streptococcus suis Isolates Recovered from Pigs in Spain"

_antibiotics, 2021, doi:10.3390/antibiotics10060707_

Round 1
Reviewer 1 Report
In their study Petrocchi-Rilo et al. provide a characterization of a collection of Streptococcus suis isolates from Spain from various sources. Although such data might be of interest for a potential reader I have several reservation about this paper:
- Introduction: the paragraph describing virulence factors needs some clarifications and editing.
- The information concerning the way the isolates were collected is currently too scarce to evaluate how representative the collection is. No data are provided on number of farms, number of samplings, type of farms, source of piglets etc.
- Authors have chosen serotyping as typing method, but this approach provides very general information on clonality. More up-to date methods should be applied (MLST or preferably genomic sequencing).
- There is no need to provide tables with sequences of primers described in other studies (tables 6, 7, 8).
- DNA isolation procedure is not described.
- Table 1: number of farms providing isolates of each serotype may be provided here.
- Tables 2, 4, 5 basically list observed features. It would be more informative to group each of these (virulence profiles, antimicrobial profiles, resistance genes) by serotypes.
- Detection of the fexA gene: Authors claim the first observation of this gene in S. suis but do not describe a positive control for PCR or sequencing of the obtained PCR product. Such a finding should be confirmed by genomic sequencing of at least one isolate.
- Discrepancy between observed chloramphenicol resistance and distribution of fexA: the cat gene should be additionally tested.
- Discussion: Should be much more concise and more focused on discussing Authors own data in the context of other findings.
- The manuscript needs correction of English usage.
Author Response
Dr. Ma,
Assistant Editor,
Antibiotics, MDPI,
May 11th, 2021
Antibiotics-1208748
Anatomical Site, Typing, Virulence Gene Profiling, Antimicrobial Susceptibility and Resistance Genes of Streptococcus suis Isolates
Dear Dr. Ma,
We are sending you the revised edition of our manuscript, in which we have taken into consideration most of the suggestions made by the reviewers. When it has been impossible to address certain comments, we include our rebuttal.
In addition, the English has been considerably improved. Even so, if you consider that the English needs improvement of style, please, tell us as soon as possible for sending our manuscript to an editing service. All changes appear in red in the revised manuscript.
Reviewer 1:
- Introduction: the paragraph describing virulence factors needs some clarifications and editing.
This paragraph has been extensively edited (lines 53 to 64).
- The information concerning the way the isolates were collected is currently too scarce…
The information has been adequately completed (lines 300-305 and Table 1), as it is suggested by this reviewer.
- Authors have chosen serotyping as typing method, but its approach provides very general information…
The typing method was a multiplex PCR, not serotyping using antisera. Even so, the term “serotype” remains being typically used. On the other hand, the purpose of our investigation was not genomic sequencing (economically unsustainable for us for more than 200 isolates), but the search of those resistance genes to those antimicrobial agents mainly used in pig farming.
- There is no need to provide tables with sequences…
Tables 6-8 have been already deleted in the revised manuscript.
- DNA isolation procedure is not described.
DNA extraction appears now in the revised manuscript (lines 299, 311-314).
- Table 1: number of farms providing…
This information is already included in the second column in Table 1.
- Tables 2, 4, 5 basically list observed features. It would be more informative to group…
The information is given now by serotypes in these tables.
- Detection of the fexA gene: Authors claim the first observation of this gene…
For the fexA gene identification, the amplification of this gen was developed. When a positive result was obtained (band size), this result remained being used as control in the next PCRs. Certainly, it would have been interesting sequencing the PCR product, but sequencing should have been done with all genes in this case. This sequencing had been unfeasible for us, both economically and because of time. Anyway, the sentence referred to the “first observation of fexA gene” has been deleted in the revised article.
- Discrepancy between observed chloramphenicol resistance and distribution of…
We think that the reviewer 1 wanted to say florfenicol instead of chloramphenicol. Anyway, the suggestion of including the possible action of cat gene, as suggested this reviewer, is added in the revised manuscript along to a reference (lines 267-269).
- Discussion: Should be much more concise…
Discussion section has been considerably shortened (about one page) and authors have focused it in the personal data.
- The manuscript need correction of English usage.
The English has been substantially improved in our revised manuscript.
Your sincerely,
César B. Gutiérrez Martín
Reviewer 2 Report
An overall interesting study on a very significant swine pathogen with some issues that need improvement. Please find attached the respective comments.

Author Response
Dr. Ma,
Assistant Editor,
Antibiotics, MDPI,
May 11th, 2021
Antibiotics-1208748
Anatomical Site, Typing, Virulence Gene Profiling, Antimicrobial Susceptibility and Resistance Genes of Streptococcus suis Isolates
Dear Dr. Ma,
We are sending you the revised edition of our manuscript, in which we have taken into consideration most of the suggestions made by the reviewers. When it has been impossible to address certain comments, we include our rebuttal.
In addition, the English has been considerably improved. Even so, if you consider that the English needs improvement of style, please, tell us as soon as possible for sending our manuscript to an editing service. All changes appear in red in the revised manuscript.
Reviewer 2:
The authors strongly thank reviewer 2 for his/her extensive revision directly in the manuscript. All his/her suggestions have been taken into consideration.
Concerning his/her issue about a study of correlation/association between metaphylaxis in farms and the resistance patterns obtained by us, although this study would have very interesting, we could not have access unfortunately to this information from the farms were tested by us.
Your sincerely,
César B. Gutiérrez Martín
Reviewer 3 Report
In this manuscript the authors investigated a collection of 207 Streptococcus suis isolates from pigs in Spain, collected between 2019 and 2020. Next to serotyping they analyzed the presence of virulence factors and resistance genes by PCR, as well as determined the inhibitory profiles of 18 antibiotics from multiple different antibiotic classes against half of the isolated strains. They found a total of 13 different serotypes and about 12% non-typeable strains, whereas three serotypes (1, 2, and 9) accounted for more than 60% of isolates in a rather equal distribution. The pattern of 5 virulence genes was very diverse and one quarter of isolates exhibited the same four virulence genes. Resistance testing and resistance gene analysis yielded an alarmingly high rate of multi-resistance strains.
This manuscript by Petrocchi-Rilo et al. is well written, although the English could be improved. The conclusions drawn are legitimate.
Minor changes
There are some spelling errors or awkward wordings which should be corrected (see pdf with correction suggestions)
In general, analyzed genes from Table 6 should be explained in the legend or methods section (not only abbreviations like cps1J or cps5N). E.g. which genes encode glycosyl-transferases, capsular polysaccharide repeat unit transporter genes etc., similar as in the methods section of Kerdsin et al. (citation 53).
In the discussion a statement about the untypeable isolates would be anticipated. It is mentioned that species identity was checked via phenotypic characterization and PCR of the gdh gene. It is not mentioned If the PCR products have been sequenced. Could non-typeable isolates also be other, closely related species?

Author Response
Dr. Ma,
Assistant Editor,
Antibiotics, MDPI,
May 11th, 2021
Antibiotics-1208748
Anatomical Site, Typing, Virulence Gene Profiling, Antimicrobial Susceptibility and Resistance Genes of Streptococcus suis Isolates
Dear Dr. Ma,
We are sending you the revised edition of our manuscript, in which we have taken into consideration most of the suggestions made by the reviewers. When it has been impossible to address certain comments, we include our rebuttal.
In addition, the English has been considerably improved. Even so, if you consider that the English needs improvement of style, please, tell us as soon as possible for sending our manuscript to an editing service. All changes appear in red in the revised manuscript.
Reviewer 3
The authors strongly thank reviewer 3 for his/her extensive revision directly in the manuscript. All his/her suggestions have been taken into consideration.
Concerning his/her question about the analyzed genes from Table 6, we have had to delete tables 6-8 at the suggestion of reviewer 1.
In relation to the statement about the untypeable isolates, a paragraph appears in the revised manuscript (lines 216-222). In addition, until three possible explanations for untypeable S. suis isolates are given. We do not think that untypeable isolates could be other closely related species because in all previous investigations about typing, always a certain percentage of untypeable isolates han been reported.
Yours sincerely,
César B. Gutiérrez Martín